# Personnel Scheduling during the COVID-19 Pandemic: A Probabilistic Graph-Based Approach

**DOI:** 10.3390/healthcare11131917

**Published:** 2023-07-03

**Authors:** Mansoor Davoodi, Ana Batista, Abhishek Senapati, Justin M. Calabrese

**Affiliations:** 1Center for Advanced Systems Understanding (CASUS), Helmholtz-Zentrum Dresden Rossendorf (HZDR), 01328 Görlitz, Germany; a.batista-german@hzdr.de (A.B.); a.senapati@hzdr.de (A.S.); j.calabrese@hzdr.de (J.M.C.); 2Department of Ecological Modelling, Helmholtz Centre for Environmental Research (UFZ), 04318 Leipzig, Germany; 3Department of Biology, University of Maryland, College Park, MD 20742, USA

**Keywords:** personnel scheduling, presence strategy, testing strategy, pandemic, COVID-19

## Abstract

Effective personnel scheduling is crucial for organizations to match workload demands. However, staff scheduling is sometimes affected by unexpected events, such as the COVID-19 pandemic, that disrupt regular operations. Limiting the number of on-site staff in the workplace together with regular testing is an effective strategy to minimize the spread of infectious diseases like COVID-19 because they spread mostly through close contact with people. Therefore, choosing the best scheduling and testing plan that satisfies the goals of the organization and prevents the virus’s spread is essential during disease outbreaks. In this paper, we formulate these challenges in the framework of two Mixed Integer Non-linear Programming (MINLP) models. The first model aims to derive optimal staff occupancy and testing strategies to minimize the risk of infection among employees, while the second is aimed only at optimal staff occupancy under a random testing strategy. To solve the problems expressed in the models, we propose a canonical genetic algorithm as well as two commercial solvers. Using both real and synthetic contact networks of employees, our results show that following the recommended occupancy and testing strategy reduces the risk of infection 25–60% under different scenarios. The minimum risk of infection can be achieved when the employees follow a planned testing strategy. Further, vaccination status and interaction rate of employees are important factors in developing scheduling strategies that minimize the risk of infection.

## 1. Introduction

Personnel scheduling decisions are crucial in many organizations since labor cost constitutes one of the major expenses in operations management. Thus, any improvement in staffing and scheduling decisions would result in overall organizational benefits [1]. Staffing and scheduling decisions can be subject to unexpected events that should be managed proactively to ensure that performance measures are met. A recent global-scale phenomenon that considerably impacts scheduling decisions is the COVID-19 pandemic [2]. During a pandemic, it is necessary to consider a hybrid work strategy to limit the number of employees present in the workplace to ensure employee safety. Another efficient strategy to mitigate the impact of a pandemic is implementing testing, which organizations may offer to their employees. Due to the limitations in the testing capacity and sensitivity, efficient applications of tests are necessary to prevent virus outbreaks in the workplace. Therefore, it is crucial to derive efficient staff scheduling and testing strategies to guarantee safety in the workplace while achieving the organization’s goals [3].

The personnel scheduling problem in pandemic situations is an emerging topic that has not been extensively addressed. The question of defining a scheduling plan that accounts for testing strategies to reduce the risk of infection while ensuring low levels of understaffing remains unanswered. Therefore, optimal scheduling, together with an optimal testing strategy, significantly contributes to keeping the workspace safe. In this paper, we aim to fill this gap by developing two Mixed Integer Non-linear Programming (MINLP) models considering a probabilistic graph-based approach to determine the optimal workplace occupancy that minimizes the risk of infection. The graph-based approach assumes that employees are in close contact with each other, which contributes to the virus’s spread. The main objective is to minimize the expected risk of infection while constraining workplace occupancy to comply with COVID-19 regulations.

Our models deal with two different realistic scenarios. The first model considers a situation where employees frequently underestimate the adherence to testing protocols. It provides both optimal personnel scheduling at the workplace and their testing strategies. On the other hand, the second model assumes a random testing strategy for the employees and derives only the optimal presence scheduling. We propose two approaches to solve the non-linear models. The first approach applies commercial optimization solvers; APOPT for the first model and Gurobi for the second model. To this end, we linearize an equation (the equation for computing and updating the probability of infection) in the models. The second approach is a canonical genetic algorithm that utilizes penalization to satisfy the constraints of the models. We consider both real contact network data of employees and randomly generated sparse and dense graphs while assessing the models’ performance under several scenarios. The results show significant impacts of both presence rate and testing schedule in minimizing the risk of infection.

This paper is organized into five sections. After reviewing related and recent studies in the following of this section, the problems of finding optimal presence and testing strategies are formulated in two different models in Section 3. Section 4 presents a heuristic algorithm for solving the models compared with the use of commercial non-linear solvers. Numerical results for different scenarios are presented in Section 5. Finally, a conclusion is drawn in Section 6.

## 2. Related Work

Personnel scheduling is one of the critical decisions in organizations; however, it is impacted by both expected events like demand or capacity uncertainty and by unexpected events like the COVID-19 pandemic. While the first kind of events usually can be handled by considering labor flexibility strategies (e.g., multiskilled staff, flexible contracts, collaborative teams) to minimize the mismatch between supply and demand [4], the second kind is difficult to handle. These events certainly affect the performance of some organizations (e.g., service sector, retail, healthcare, manufacturing), which must continue with regular operations despite the global health crisis.

Extensive literature on personnel scheduling problems exists [1,5,6,7]. According to the classification defined in [7], we can categorize this study as disruption management, in which the aim is to derive robust schedules by reducing the impact of the effects caused by a health emergency, such as a pandemic.

The personnel scheduling problem in a pandemic situation is an emerging topic that largely started with the COVID-19 pandemic [2,8,9]. In contrast to existing studies on disruption management [10,11,12,13,14] that focus mainly on developing strategies to cope with staffing operational disruptions (e.g., demand variations, airline crew delays, nurse absenteeism), this problem concerns employees’ health and safety, requiring additional considerations, such as the control of the virus spread among the staff while satisfying staffing levels. The existing studies in the literature are focused on developing scheduling policies to prevent the spread of the virus in organizations and closed spaces. For residential care facilities, ref. [15] developed a task scheduling model to minimize the number of employees assigned to residents to control the spread of the virus. To solve the model, the authors proposed a population-based heuristic algorithm that guarantees solution quality against benchmark solution approaches. Ref. [16], studied the problem of scheduling physicians during the COVID-19 pandemic in a hospital in Turkey. The authors proposed a Mixed Integer Programming (MIP) model to solve a shift scheduling problem to guarantee the safety of the physicians while keeping a balanced workload in the hospital.

During the COVID-19 pandemic, demand for hospital care often exceeded supply. Gao et al. [17] studied a Medical Staff Rebalancing (MSR) problem to allocate medical staff to different areas, considering the demand as the number of infected patients in the allocation regions. To address the MSR problem, the authors proposed two robust optimization models that account for uncertainty in data availability while ensuring allocation fairness during emergencies. Similarly, ref. [18] proposed a scheduling model to minimize the workload unbalance of the nurses in charge of COVID-19 patients. To solve this problem, they developed a Hybrid Salp Swarm Algorithm and Genetic Algorithm (HSSAGA) and showed their algorithm outperformed the state-of-the-art solution approaches.

For a pharmaceutical distribution warehouse in Italy, ref. [19] developed a Mixed Integer Linear Programming (MILP) model to solve a shift scheduling problem. The aim was to minimize the deviation in allocated contractual hours of employees during the COVID-19 pandemic to keep operations ongoing while guaranteeing the safety of the employees. Guerriero and Guido [20] proposed a flexible staff scheduling approach for a University administrative department during the pandemic. By allowing a hybrid work system, they developed a days-off optimization model considering employee preferences and availability. Alwadood et al. [21] considered the personnel scheduling problem for a hotel housekeeping department used as a quarantine center for foreign travelers. This study proposed a weekly schedule for the staff using a Binary Integer Programming model that minimizes the workforce on duty to decrease the risk of infection.

## 3. Modeling a Graph-Based Personnel Scheduling Problem during Pandemic Situations

Personnel scheduling during a pandemic requires special handling to ensure employee safety while continuing regular operations. Since the virus that spreads COVID-19, the Severe Acute Respiratory Syndrome Coronavirus 2 (SARS-CoV-2), is transmitted through individual contacts [22], the World Health Organization [23] suggested several measures to control the spread of the virus, including social distancing, testing, and vaccination.

This section proposes two MINLP models to solve the personnel scheduling problem during disease outbreaks, particularly the COVID-19 pandemic. We aim to derive a *presence strategy* to find the optimal schedule of employees (i.e., working remotely or at the workplace) and a *testing strategy* to determine the testing days of the employees. We consider an organization with *n* employees who are in contact with each other and have to be allocated in a discrete-time horizon, d=1,2,…,D (i.e., a week, D=5) such that the risk of infection in the workplace is minimized. The proposed models compute the probability of infection for each employee under two different cases. The first model assumes that the employees comply with the testing protocols following the suggested testing days. The second model does not impose strict regulations on testing, so the employees perform tests arbitrarily during the evaluated time horizon. The notation and assumptions considered in the proposed models are listed below.

**Assumptions**:
The tests are performed in the morning before employees come to the workplace, and if the result is positive, they stay at home.We initialize the probability of infections to background risk, i.e., $PI_{i}^{0}=br$, for i=1,2,…,n. Indeed, we assume for the starting day of the scheduling interval (i.e., Monday), the employees’ risk is the same as the background risk in the organization’s neighborhood.We assume working from home is free of risk for the employees, i.e., if the employees $e_{i}$ do not come to the workplace on the day *i*, $PI_{i}^{d}=PI_{i}^{d-1}$. However, this assumption can be easily relaxed by applying the risk probability to the related computations.

### 3.1. Computing the Probability of Infection in a Graph-Based Approach

In this subsection, we propose a graph-based approach to compute the probability of infection in the workplace. Let $PI_{i}^{d}$ be the probability of infection for $e_{i}$ at the end of the working day *d*, for i=1,2,…,n and d=1,2,…,D. The objective function is to minimize the expected risk of infection inside the facilities, which is directly related to minimizing the probability of infection of employees. Thus,
(1)$$Minimize\;\;Z=\frac{1}{D\times n}\;\;\sum_{d=1}^{D}\sum_{i=1}^{n}PI_{i}^{d}.$$

To compute the probability $PI_{i}^{d}$, we present a two-step procedure. In the first step, we calculate an intermediate probability ($PI_{i}^{{}^{\prime }d}$) based on a recursive formula that incorporates the presence and testing strategies for the employees. As mentioned in the assumptions, the initial probability of infection $PI_{0}^{i}$ is set to br, which means a risk of infection based on the number of incidences reported in the organization’s neighborhood. In the second step, we calculate the final probability by considering the graph of connectivity among the employees.

**Step 1**: If an employee with a probability of infection *p* performs a test and the result is negative, the infection probability will reduce to p×FN, where FN is the probability of false negativity for the tests. Now, let the binary variable $t_{i}^{d}$ indicate $e_{i}$ is tested on day *d* before coming to the workplace (i.e., $t_{i}^{d}=1$) or not (i.e., $t_{i}^{d}=0$). So, the updating recursive formula can be written as
(2)$$PI_{i}^{{}^{\prime }d}=PI_{i}^{d-1}\times \left(1-t_{i}^{d}\right)+PI_{i}^{d-1}\times t_{i}^{d}\times FN.$$

Equation (Equation 2) works well when the employees follow the testing strategy. If they do not follow this strategy, we can apply a random testing strategy, that is, assuming a testing probability pr(test) for each employee. For example, if the employees take two tests in five days, the Probability of testing per day is $pr\left(test\right)=\frac{2}{5}$. If $e_{i}$ performs a test at day *d* with probability pr(test), then we can use the following equation for applying the effect of tests and computing $PI_{i}^{{}^{\prime }d}$.
(3)$$PI_{i}^{{}^{\prime }d}=\left(1-pr\left(test\right)\right)\times PI_{i}^{d-1}+pr\left(test\right)\times PI_{i}^{d-1}\times FN.$$

Therefore, regarding the fact that the employees follow a recommended testing strategy, Equation (Equation 2), or a random test strategy, Equation (Equation 3), we apply the effect of performing the tests and update the probability of infection for all employees. Then, we update the probabilities of infection for the employees based on their contacts.

**Step 2:** If $e_{i}$ comes to the workplace on day *d* (i.e. $x_{i}^{d}=1$), and with the probability of $p_{ij}$ contacts with $e_{j}$ (suppose $x_{j}^{d}=1$ as well), then the probability of infection for him/her can be updated as:(4)$$PI_{i\leftarrow j}^{d}=1-\left[\left(1-PI_{i}^{{}^{\prime }d}\right)\times \left(1-p_{ij}\times \beta_{i}\times PI_{j}^{{}^{\prime }d}\right)\right],$$
where $\beta_{i}$ is the probability of infection for $e_{i}$ in the case that $e_{j}$ is infected. For the sake of simplicity, we only assume two possible values for $\beta_{i}$, whether the employee is vaccinated or not. $PI_{i\leftarrow j}^{d}$ denotes the effect of contact with $e_{j}$. Thus, by applying all possible contacts $e_{i}$ may have during day *d*, the infection probability at the end of the day can be computed as follows:(5)$$PI_{i}^{d}=1-\left[\left(1-PI_{i}^{{}^{\prime }d}\right)\times \prod_{j=1\&j\neq i}^{n}\left(1-p_{ij}\times \beta_{i}\times x_{j}^{d}\times PI_{j}^{{}^{\prime }d}\right)\right].$$

For the sake of simplicity, let us denote the two-step procedure by a function based on the related parameters if the employees follow the recommended testing strategy:(6)$$PI_{}^{d}=Update\left(PI_{}^{d-1},x_{}^{d},t_{}^{d}\right),$$
and, if they follow a random testing strategy:(7)$$PI_{}^{d}=Update\left(PI_{}^{d-1},x_{}^{d},pr\left(test\right)\right),$$
where $PI_{}^{d-1}=\left\{PI_{1}^{d-1},\;PI_{2}^{d-1},\;\ldots ,\;PI_{n}^{d-1}\right\}$ is the infection probability for the employees, and $x_{}^{d}=\left\{x_{1}^{d-1},\;x_{2}^{d-1},\;\ldots ,\;x_{n}^{d-1}\right\}$ and $t_{}^{d}=\left\{t_{1}^{d-1},\;t_{2}^{d-1},\;\ldots ,\;t_{n}^{d-1}\right\}$ are the presence indicator and testing indicators for all the employees in the day *d*, respectively.

### 3.2. Personnel Scheduling and Testing Strategy Models

In this section, we define two MINLP models considering the probability-of-infection equations defined in Section 3.1. Model 1 assumes that the employees follow a recommended testing strategy, and Model 2 is based on the fact that employees do not follow the testing protocols. Thus, they test themselves randomly with a probability of pr(test) for each day. We also consider a set of constraints. We keep the model simple and easy to understand. In practice, different organizations may have their specific limits and constraints, which can be added to the model. Here, we consider two families of upper and lower bound constraints: the first family of constraints is related to satisfying the on-site tasks in the organization. They are defined as:(8)$$\sum_{i\in C_{k}}^{n}x_{i}^{d}\geq b_{k},\;\;for\;\;k=1,2,\ldots ,m,$$
where $C_{k}$ is the k-th subset of employees such that at least $b_{k}$ number of them have to present at the workplace on day *d*. As an example, assume the task of registration and deregistration of citizens in the municipality. In the usual situation, it may need, for example, five employees present at office in order to carry out the related tasks. However, in the pandemic, the municipality may reduce the minimum necessary employees to two employees per day. The second family of constraints refers to capacity limitations in the workplace. In fact, during the COVID-19 pandemic, there were regulations on the maximum number of employees who could be simultaneously (in a day) present in the workplace. As an example, assume there is an office that usually has five employees on staff where, due to the pandemic and the maintenance of safe social distance, only three of them can be working simultaneously. So, we can model them as follows:(9)$$\sum_{i\in C_{k}^{\prime }}^{n}x_{i}^{d}\leq b_{k}^{\prime },\;\;for\;\;k=1,\;2,\;\ldots ,\;m^{\prime },$$

Note that in real-world applications, scheduling can be conducted for short intervals, such as a week, i.e., D=5 working days. This is essential due to a higher rate of change in the incidence level, which plays a basic role in initializing background risk $PI^{0}$. As a result, in order to keep the model straightforward, we do not include the time-related constraints that may apply to lengthy scheduling intervals.

Model 1 would result in a lower risk of infection compared to Model 2, but it requires the employees to follow the recommended testing strategy. In contrast, Model 2 depicts a more flexible testing scheme, in which the employees apply the offered tests randomly during the scheduling period. The models are defined as follows:


**Model 1: Personnel Scheduling with Testing Strategy**




(10)
$$\begin{array}{cc} \; & Minimize\;\;\;Z=\frac{1}{D\times n}\;\;\sum_{d=1}^{D}\sum_{i=1}^{n}PI_{i}^{d},\\ \; & \;\;\;\;Subject\;to:\\ \; & \;\;\;\;\;\;\;\;\;\;\;\;\;PI_{}^{0}=\left\{\beta_{1}br,\beta_{2}br,\ldots ,\beta_{n}br\right\}\\ \; & \;\;\;\;\;\;\;\;\;\;\;\;\;PI_{}^{d}=Update\left(PI_{}^{d-1},x_{}^{d},t_{}^{d}\right),\;\;for\;d=1,2,\ldots ,D,\\ \; & \;\;\;\;\;\;\;\;\;\;\;\;\sum_{i\in C_{k}}^{n}x_{i}^{d}\geq b_{k},\;\;for\;\;k=1,2,\ldots ,m,\\ \; & \;\;\;\;\;\;\;\;\;\;\;\;\sum_{i\in C_{k}^{\prime }}^{n}x_{i}^{d}\leq b_{k}^{\prime },\;\;for\;\;k=1,2,\ldots ,m^{\prime },\\ \; & \;\;\;\;\;\;\;\;\;\;\;\;\sum_{d=1}^{D}t_{i}^{d}\leq TC_{i},\;\;for\;\;i=1,2,\ldots ,n,\\ \; & \;\;\;\;\;\;\;\;\;\;\;\;x_{i}^{d},t_{i}^{d}\in \left\{0,1\right\},\;\;for\;i=1,2,\ldots ,n,\;\;and\;\;d=1,2,\ldots ,D \end{array}$$



$TC_{i}$ refers to the test capacity for $e_{i}$; the maximum number of available test kits for $e_{i}$ in a period of *D* days. This capacity may be the same for all employees or distributed among the employees according to the number of connections each employee has. Model 1 has two decision variables, presence scheduling, $x_{i}^{d}$, and testing schedule, $t_{i}^{d}$, for i=1,2,…,n and d=1,2,…,D. So, the model will (optimally) derive which employees to allocate in the workplace and when, and on which days to perform the tests. If in an organization, the employees do not follow the suggested testing strategy, and they use the tests arbitrarily during the scheduling period, Model 1 will not fit with that organization. In this case, the following model, which assumes the tests can be used by the employees with a probability, is a better match for that organization.


**Model 2: Personnel Scheduling without Testing Strategy**




(11)
$$\begin{array}{cc} \; & Minimize\;\;\;Z=\frac{1}{D\times n}\;\;\sum_{d=1}^{D}\sum_{i=1}^{n}PI_{i}^{d},\\ \; & \;\;\;\;Subject\;to:\\ \; & \;\;\;\;\;\;\;\;\;\;\;\;\;PI_{}^{0}=\left\{\beta_{1}br,\beta_{2}br,\ldots ,\beta_{n}br\right\}\\ \; & \;\;\;\;\;\;\;\;\;\;\;\;\;PI_{}^{d}=Update\left(PI_{}^{d-1},x_{}^{d},pr\left(test\right)\right),\;\;for\;d=1,2,\ldots ,D,\\ \; & \;\;\;\;\;\;\;\;\;\;\;\;\sum_{i\in C_{k}}^{n}x_{i}^{d}\geq b_{k},\;\;for\;\;k=1,2,\ldots ,m,\\ \; & \;\;\;\;\;\;\;\;\;\;\;\;\sum_{i\in C_{k}^{\prime }}^{n}x_{i}^{d}\leq b_{k}^{\prime },\;\;for\;\;k=1,2,\ldots ,m^{\prime },\\ \; & \;\;\;\;\;\;\;\;\;\;\;\;x_{i}^{d}\in \left\{0,1\right\},\;\;for\;i=1,2,\ldots ,n,\;\;and\;\;d=1,2,\ldots ,D \end{array}$$



Both Model 1 and Model 2 are MINLP and, like the general scheduling problem [24] with hard constraints, are NP-hard. Furthermore, considering the upper bound and lower bound constraints of the problem, finding even a feasible solution that satisfies the constraints is an intractable problem. The main difficulty of the models is updating the risk of infection for the employees after daily contacts, Equation (Equation 5). It is an exponential equation and impossible for most algorithms to cope with. Therefore, in the following, we present an efficient simplification to handle this issue.

It is worthwhile to mention, both models require the contact network among the employees. That means the (average) contact rate between any pair of employees should be known. Further, the models assume a risk of infection for the employee when they work from home, which may differ in practice. Indeed, the infection risk varies from employee to employee, depending on their connection with family members and friends. Finally, the models do not consider the absence of employees, which may result from sick leaves. This works well when the rate of sick leaves does not affect workflows, and if it is not the case, the constraints on the minimum number of on-site employees should be satisfied with a confident threshold. **Relaxation** The term $\prod_{j=1\&j\neq i}^{n}\left(1-p_{ij}\times \beta_{i}\times x_{j}^{d}\times PI_{j}^{{}^{\prime }d}\right)$ in Equation (Equation 5) is the only exponential equation of the proposed models. As explained, this term is used for updating the infection risk of an employee after his/her daily contacts. In practice, the value of this term is so small (based on the data and experiments, that it is on the order of $10^{-5}$. So, to relax the models and remove this exponential term, we use the linear Taylor expansion of the formula, ${\left(1-x\right)}^{n}\approx 1-nx$. Thus, the simplified approximation of Equation (Equation 5) can be written as below:(12)$$PI_{i}^{d}=1-\left[\left(1-PI_{i}^{{}^{\prime }d}\right)\times \left(1-\sum_{j=1\&j\neq i}^{n}\left(p_{ij}\times \beta_{i}\times x_{j}^{d}\times PI_{j}^{{}^{\prime }d}\right)\right)\right],$$

To evaluate the accuracy of this simplification, we compared the above equation with Equation (Equation 5) using the parameters reported in the example presented in Section 5.1. The comparison showed that the equations result in almost the same values, with a precision on the order of $10^{-9}$ on average. Thus, it is a suitable linear approximation in practice.

## 4. Solution Approach

To solve the proposed Model 1 and Model 2, we developed different solution approaches. For each model, we apply a nonlinear commercial solver; GEKKO and APOPT (v1.0) [25] solver for the first model, and Gurobi 5.6.3 optimization solver [26] for the second model. To apply these solvers, we replace the Equation (Equation 5) with linearization explained in Equation (Equation 12). In addition to applying these solvers, we propose a Genetic Algorithm (GA) and tailored it for both Model 1 and Model 2. In the following, we briefly explain the GA and its operators.

Genetic algorithms (GAs) are random search algorithms which work based on heuristic *exploration* and *exploitation* operations [27]. The GA starts with a random set of solutions (chromosomes), called the *population*. Then, it evolves the population generation by generation, using some exploration and exploitation operators. To this end, the *selection* operator chooses some high-fitness solutions as the *parent* chromosomes and puts them in the *mating pool*. Then, the *Crossover* operator takes a pair of such parents and produces (usually) two new chromosomes, i.e., the *children*. Indeed, it combines one part (some *genes*) from the first parent with the other part of the second parent and vice versa. The *mutation* operator mimics the natural mutation and changes some genes of a child solution at random to explore a new search space and prevent the (premature) convergence of the population. Both of the operators play the role of exploration. Finally, at the end of each iteration, from the combined parent and children populations, a set of high-fitness chromosomes are picked for the next generations.

There are several kinds of crossover, mutation and selection operators [27,28]. We present a canonical GA with standard tournament selection operators, single-point crossover, and swap-mutation operators. Since the decision variables are binary, we directly use them to represent a chromosome. Also, to satisfy the constraints of the models, we penalize the infeasible chromosomes by adding a penalty value, that is, the number of violations of the constraints that occurs. Finally, we define the fitness function as the sum of the objective function, the expected risk of infection defined in Equation (Equation 1), and the penalty value. Note that the objective function is always a value between zero and one. So, this definition of the fitness function emphasizes the priority of finding feasible solutions in the search space, followed by improving the solutions in terms of the risk value. Therefore, the GA can also be applied to finding a feasible solution, such as the first solution with a penalty value of zero found in the population. We utilize the proposed GA in two folds, finding a random feasible solution that satisfies all the constraints, and a (feasible) suboptimal solution that minimizes the expected risk of infection among the employees. We use such random solutions in comparing and showing the impact of the presence and testing strategies.

We applied GA to solve the proposed Model 1 and Model 2. Its complexity depends on the size of its population and the number of generations, and there is a trade-off between the complexity and the optimality of the obtained solutions. On the other hand, it is straightforward to apply the GA on either Equation (Equation 5) or its relaxed linear equation, Equation (Equation 12). For *n* employees and a period of *D* days, any chromosome can be evaluated in O(n×D×M) time, where *M* is the size of all constraints in the problem. Therefore, the time complexity of a GA with population size *p* in *g* iterations is O(p×g×n×D×M) time. Thus, all the parameters have a linear impact on the algorithm’s complexity. Finally, GA is a random search, meaning multiple runs of it on the same instance of a problem may result in different solutions.

In contrast to the GA, the time complexity of solvers Gurobi and APOPT in GEKKO depends on the number of employees and the size of constraints, which have an exponential impact on the complexity of such solvers in the worst case. Therefore, these solvers use different approaches to avoid the long-running time, such as solving the problem in the dual space and applying predetermined optimality gaps, pre-solving, and branch and bound techniques. We applied APOPT to Model 1 and Gurobi to Model 2 with the linear Equation (Equation 12). Although the running time of these solvers depends on the whole of the parameters in the models, they are faster than the GA when a logical optimality gap size like 10${}^{-5}$ is used. However, the objective value of solutions obtained by the GA is better than those obtained by the solvers. In Section 5.4, we evaluate the performance of the different solution approaches in terms of running time and solution optimality by comparing the APOPT solver and GA for Model 1 and the Gurobi solver and GA for Model 2.

## 5. Numerical Results

In this section, we evaluate the proposed models and algorithms on several test problems. The aim is to find the optimal presence and testing strategies that result in the minimum expected risk of infection among the employees. We compare the models and the algorithms separately. Further, we analyze the impact and sensitivity of the models and parameters. The experiments are composed of four parts. In the first part, we assume a small-size organization and show the optimal presence and testing strategies. In the second part, we consider real data on employee contact networks in organizations and random connectivity graphs. In the third part, the results for random connectivity graphs are represented, and finally, the fourth part compares the effectiveness of the algorithms. Since in the first three parts, the aim is to compare the models and impact of the introduced parameters, we run the presented genetic algorithms 30 times and report the average objective’s values. We consider the following values for the model’s parameters.

We consider a time horizon of a week, e.g., five working days, D=5 for running the experiments.We define the probability of disease transmission as β=0.1. We choose this value as a probable pessimistic case from a possible range of values reported in previous studies [29,30,31,32] regarding the first variants of COVID-19 (*Alpha*, *Delta*, and *Omicron*, which is more transmissible than the previous ones).We calculate the background risk based on seven-day incidences of COVID-19 infections in Saxony, Germany, in the period of June–August 2022. We consider 300 incidences on average for this period and set the daily background risk to $br=\frac{1}{7}\times \frac{300}{100,000}$.The initial risk of infection, $PI^{0}$, e.g., the risk of infection on Mondays, is determined by applying the background risk and two-day weekend. That is, $PI^{0}=1-{\left(1-br\right)}^{2}$.We apply the impact of vaccine (fully vaccinated, that is, at least two doses of a COVID-19 vaccine have been received) on the initial risk of infections and transmission probability during the employee’s interaction. Based on the related recent research (e.g., see [33,34,35,36]), on average, we set the transmission rate of vaccinated employees to (1−0.85)β, which implies 85% immunity for them.

### 5.1. Base Case Study

In this subsection, we present the base case study in which we consider a small-size organization with 20 employees. The organization is divided into two cross-functional sections: the first section includes 12 employees, i.e., $e_{1},e_{2},\ldots ,e_{12}$, and the second one includes $e_{13},e_{14},\ldots ,e_{20}$. The connectivity graph that represents the employee’s contact network is shown in Figure 1. We consider the following allocation rules on the presence of the employees:(i)Each employee has to be present at the workplace at least two days a week,(ii)The whole workplace occupancy should remain between MinOccupation=50% and MaxOccupancy=75% of all the employees. That means at least 10 and at most 15 employees can be present at the workplace daily.(iii)At least 30% of employees in each section should be present at the workplace. That means, at least four employees from the first section and three from the second section should be present daily at the workplace.

In addition, we assume that the probability of false negativity of the tests is FN=0.2, and two tests are available per employee per week. That means, in the second model, the probability of testing per (working) day is $pr\left(test\right)=\frac{2}{5}=0.4$, and in the first model, TC=2. Table 1 and Table 2 show the results obtained for Models Equation 10 and Equation 11. As can be extracted from the results, the presence strategy aims to satisfy the occupancy constraints by selecting employees who have a weak connection rate with each other for onsite work. Further, the strategy satisfies the minimum 50% occupancy with exactly 10 employees every day, except with a strategy of 11 employees on Thursdays in the second model. Also, in the results obtained for the second model, the testing strategy suggests employees apply the tests (two tests are available per employee) on the first days of the week. This is because we initialize the risk of infection on Mondays with a higher value after two days of the weekend. So, the strategy tries to reduce this value by suggesting tests before the employees are in contact with each other.

The expected risk of infection for the suggested strategies is $4.61\times 10^{-5}$ for the first model and $1.99\times 10^{-5}$ for the second model. That means if the employees follow the suggested testing strategy, they can reduce the risk to 43% of the risk when they follow a random testing strategy.

It may also be interesting to see what happens if the employees perform one test or three tests per week. To answer this question, we run the models for TC= 1 and 3, and pr(test)= 0.2 and 0.6. We skip reporting the strategies and only focus on the objective value. The risk of infection for TC= 1, and TC= 3, are $2.44\times 10^{-5}$ and $1.85\times 10^{-5}$, respectively, and the risk of infection for the second model where the employees follow a random testing strategy with probabilities pr(test)= 0.2 and pr(test)= 0.6 are $6.05\times 10^{-5}$ and $3.85\times 10^{-5}$, respectively. Having some high-level information and the price of each test, a manager can optimally decide how many tests are better to offer to the employees based on the testing strategy and the model that may fit the organization.

### 5.2. Results on Real Data

We consider publicly available face-to-face interaction data collected by the SocioPatterns collaboration [37]. This dataset contains the interactions among 92 employees recorded in 20-s intervals in an office building in France from the 24 June to the 3 July 2013 [38]. See Figure 2. Since the probability of contact for the employees is not explicitly reported in the available data, we pre-process the data to adapt the dataset to our proposed models. For employee *i* and employee *j*, we first aggregated the contacts between them over the above-mentioned period and calculated the average number of contacts per day ($c_{ij}$). Next, for each employee *i*, we divided the total number of contacts made by that employee per day by the number of colleagues he/she has and calculated the normalized average vertex degree ($d_{i}$) per day. Here, we defined employee *i* and employee *j* as colleagues if they have at least one contact over this period. Finally, we calculated the probability of contacts between *i* and *j* as follows:$$p_{ij}=\left\{\begin{array}{cc} 1, & if\;\;\frac{c_{ij}}{d_{i}}\geq 1\;\;or\;\;\frac{c_{ij}}{d_{j}}\geq 1\\ \max\{\frac{c_{ij}}{d_{i}},\frac{c_{ij}}{d_{j}}\} & if\;\;\frac{c_{ij}}{d_{i}}<1\;\;and\;\;\frac{c_{ij}}{d_{j}}<1. \end{array}\right.$$

We assume 95% of employees are fully vaccinated, and the probability of false negativity for the tests is FN=0.2 on average. Note that, false negativity does not have a specific rate, and research related to it has shown a wide range of values [39,40]. Under such conditions, we run the models for different scenarios, as follows:(1)Each employee has to be present at the workplace at least (i) two days in a week, and (ii) three days in a week,(2)The feasible minimum and maximum occupancy are set to (i) [30%,70%] and (ii) [40%,80%],(3)The number of available tests per employee per week is TC=1,2,3 for the first model and correspondingly pr(test)=0.2,0.4,0.6 for the second model.

Therefore, we will have 2×2×3=12 different scenarios. We run both the models under these scenarios and report the results in Table 3. We scale the risk’s values by $10^{5}$. Columns M1 and M2 show the expected risk of infection for the first and the second model, respectively. Also, to compare the impact of the models, we compute random feasible solutions (the solutions that satisfy all the constraints of the problem) for the presence and testing of the employees. The column *R* in the table shows the expected risk of infection for such solutions. This value is the average risk of infection for 30 different random solutions. As expected, the risk of infection improves from the random strategy to the strategy in the second model, and also from the second model to the first model. On average, following the suggested presence strategy can reduce the risk of infection 26% compared to the random strategy. Moreover, if the employees follow both the suggested presence and testing strategies, the risk of infection can be reduced to 60% of the random strategy, and 45% of the risk compared to the suggested strategy in the second model.

Thus, efficiently using the tests together with the presence strategy can significantly reduce the expected average risk among the employees. Also, when the tests are used randomly by the employees, in columns *R* and M2, increasing the test capacity, TC, has an almost linear impact on the risk of infection. Moreover, if the employees follow the suggested testing strategy, M1, the infection risk considerably reduces when TC increases from one to two compared to when it increases from two to three. So, the managers can offer an optimal number of free tests per week to the employees. This can be optimally chosen regarding the price of test kits and the acceptable level of risk of infection. Note that the background risk can be used as a reference point to determine such a level, and the minimum and the maximum occupation are determined by considering workplace capacities and COVID-19 regulations, work fellows, and the type of services the employees provide.

We assumed two different cases where the employees have to be present at least two days a week and where they have to be present at least three days a week. As expected, the risk of infection for the first case is less than in the second case. This constraint forces all employees to be present at the workplace, even those who are at higher risk of infection (e.g., the unvaccinated employees and employees with a higher degree of connection). The other family of constraints, which we applied to these real data, is two general constraints addressing the whole number of on-site employees per day. We considered two scenarios [30%,70%] and [40%,80%], and the results are reported separately. Despite the first family of constraints, this type of constraint allows flexibility in choosing low-risk employees to satisfy the minimum necessary number of on-site employees per day. The optimal solution (the solution which minimizes the risk of infection) usually belongs to the boundary of these constraints. Therefore, the strategy obtained for [30%,70%] is better than for [40%,80%].

### 5.3. Results on Random Graphs

In this part, we evaluate the models on randomly generated connectivity graphs. To this end, we first choose a set of nodes (each node represents an employee), and then, for any pair of nodes *i* and *j*, we assume a weight in $p_{ij}\in \left[0,1\right]$ as the probability of contact between them. We generated three small (with n=40 nodes), medium (with n=100 nodes) and large (with n=250 nodes) size graphs with two sparse and dense connections. In the sparse graphs, we set $p_{ij}$ to 1 with probability 0.05, to 0.5 with probability 0.1, and to 0 with probability 0.85. In the dense graphs, $p_{ij}=1$ with probability of 0.1, $p_{ij}=0.5$ with probability of 0.2, and $p_{ij}=0$ with probability of 0.7. We also assume three possible cases TC=1,2 and 3 for the number of available tests per employee per week, as well as two scenarios FN=0.1 and 0.3 as the probability of false negativity for each test. So, in general, there are 3×3×2=18 scenarios for the size of graphs and the test’s parameters. We considered the constraints that each employee should present at the workplace at least three days a week and the daily occupancy in the workplace is allowed to be between 50% and 75%. Table 4 and Table 5 show the risk of infection obtained for 18 sparse graphs and 18 dense graphs, respectively. The reported risk values are scaled in $10^{5}$ and they are the average of 30 independent runs.

The results in each table show the impact of the number of tests and their sensitivity, as well as following a random or suggested presence testing strategy by the employees. For example, in sparse and small-size graphs (n=40), when the employees use only one test per week (TC=1) and present at the workplace by a random strategy and apply the tests randomly on a day of the week (column *R*) if the tests have a false negativity rate FN=0.3, the results show an expected average risk of infection of $9.69\times 10^{-5}$, while for the case FN=0.1, it results in $8.19\times 10^{-5}$. These risk values can be reduced to $7.36\times 10^{-5}$ and $6.98\times 10^{-5}$ when the employees follow the suggested presence strategy with a random test strategy (column M2). Finally, following both suggested presence and testing strategies (column M1) results in $4.31\times 10^{-5}$ and $2.85\times 10^{-5}$, respectively. The table shows the results for two (TC=2) and three (TC=3) tests per week. For example, two tests with accepting suggested presence and testing strategies result in a risk of infection $3.17\times 10^{-5}$ and $1.91\times 10^{-5}$ for FN=0.3 and FN=0.1, respectively. That means 56% and 70% improvements (risk reduction) comparing the case when they do not follow the strategies. Thus, by comparing such results and the model which is more fit for the organization, an optimal case can be chosen. The same results are reported for the dense graphs, and the managers may be interested in observing the impact of the rate of contact among the employees on the expected risk of infection. In our experiments, the probability of contact between any pair of employees in the dense graphs is two times more than in the sparse graphs. However, the average risk of infection in columns R, M2 and M1 increased by factors 1.54, 1.49 and 1.41, respectively. This kind of information can be efficiently used in establishing contact regulations in the workplace space. In fact, in addition to the presence strategy and testing strategy, the rate of contact is an influential factor to control the risk of infection.

### 5.4. Comparison of Solution Algorithms

As explained, we presented a GA and applied it to both models. Further, we solve the problem defined in the first model, Model Equation 10, using GEKKO and APOPT solver, and the problem defined in the second model, Model Equation 11, using Gurobi solver. In this part, we compare these algorithms in terms of running time and efficiency in finding the optimal solution. Note that, since the problem is an NP-hard problem, none of the algorithms can guarantee to find the optimal solution in polynomial time. Thus, we compare the obtained solutions with the algorithms.

The models and algorithms were implemented in Python 3.7 on a standard PC (Intel (R) Core(TM) i7 and 32G RAM). For a graph with a specific number of nodes, we run the algorithms for 30 different sets of edges and weights and report the average running time (in seconds) of the algorithms as well as the best, mean, and worst of the objective value in the obtained solutions. We consider graphs with the number of nodes n=10,20,40,100 with different sparse and dense topologies and run the algorithms for a scheduling period D=5. We run the GA with populationsize=100+2×n, maximumgeneration=200+2×n, and mutation probability $\frac{1}{n}$. Table 6 shows the result for GA and GEKKO with APOPT solver for Model 1, and Table 7 shows the result for GA and Gurobi for Model 2.

From Table 6, it is clear the APOPT solver reaches the solutions significantly quicker than GA. However, there is always a trade-off between the running time of GA and the efficiency of the obtained solutions. APOPT surpasses GA in terms of running time. On the other hand, the GA slightly outperforms APOPT in terms of optimality. In general, both approaches can be used for Model 1. From Table 7, it is obvious Gurobi solver is much faster than GA. On the other hand, the GA outperforms Gurobi in terms of optimality. We ran Gurobi for very large graphs, e.g., with 1000 nodes, and observed that it is still efficient and can compute the solution in less than 10 s. In general, since the running time of GA is acceptable for real-size instances of the problem, it seems that both approaches can be applied to Model 2.

## 6. Conclusions and Future Work

During the COVID-19 pandemic, the most efficient strategy to prevent the spread of the virus was (and is) the implementation of teleworking and regular testing of the employees. However, to date, there is not a clear understanding of how to define efficient personnel scheduling plans while considering testing capacities so as to guarantee the safety of the employees that should keep working despite a pandemic.

This paper tackled such a situation by developing two MINLP models for deriving efficient scheduling plans during a pandemic, taking into account teleworking strategies and testing capacities. The main objective is to minimize the expected average risk of infection among the employees, considering work flow constraints and occupancy limitations in the workplace to comply with the COVID-19 regulations. The first model focuses on scheduling the presence of employees in the workplaces as well as scheduling their tests. However, since, in practice, the employee may not follow a testing schedule, we presented a second model, which aims to optimize the presence scheduling under a random testing strategy. We compared the models under various scenarios and discovered that by implementing these strategies, an organization can reduce the risk of infection by 25% to 60%. Further, we performed a sensitivity analysis by tuning several influential parameters and showed several scenarios that can significantly help managers in establishing regulations in the workplace. For instance, they can see the impact of connection weights, when it changes from dense graphs to sparse graphs, or when different occupation constraints are considered. In general, based on the simulation results and the comparison between the models, it is highly recommended for employees to adhere to the optimal scheduling extracted from the models and to also follow the suggested testing strategy. By implementing both of these measures, the highest level of occupational safety can be achieved, resulting in an average risk of infection that is reduced to half compared to when employees follow a random testing strategy.

The models we proposed assume that the number of available tests for all the employees is the same, while the tests should be distributed according to the degree of connection for the employees and the number of days on which they should be present at the workplace. The models can be easily extended to cover this case. However, by modeling a variable test availability, the search space of the problem may increase, which may require more efficient solution approaches such as branch and bound and hybrid methods.

## Figures and Tables

**Figure 1 healthcare-11-01917-f001:**
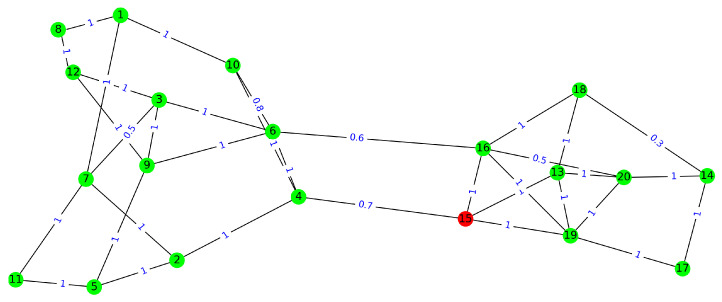
Base case study contact network: An organization with 20 employees in two cross-functional sections. All the employees except number 15 are fully vaccinated. The edge’s label shows the probability of the occurrence of contact between a pair of employees.

**Figure 2 healthcare-11-01917-f002:**
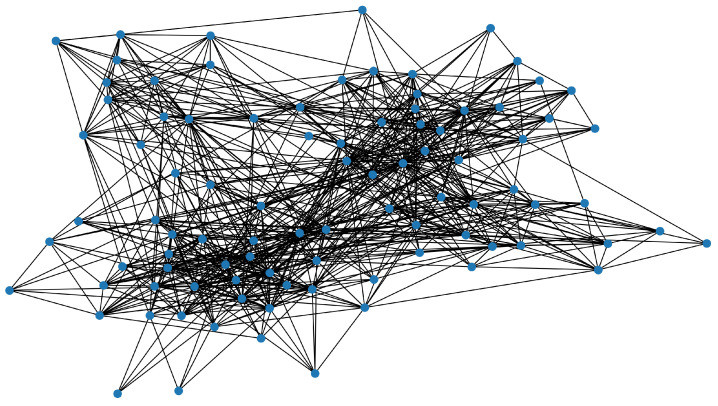
Graph representation of employees contact network with 92 employees.

**Table 1 healthcare-11-01917-t001:** Presence strategy obtained for the example illustrated in Figure 1. The number 1 indicates that the employee is present at the workplace, and 0 means working from home. The results are shown for five working days for 20 employees. In this scenario, each employee performs a test with a probability of 0.4 per day. This presence strategy results in the expected risk of infection $4.61\times 10^{-5}$.

Employee	Monday	Tuesday	Wednesday	Thursday	Friday
1	1	0	1	1	0
2	1	0	1	1	0
3	0	1	0	0	1
4	1	0	0	0	1
5	0	1	0	0	1
6	0	1	0	0	1
7	0	1	1	1	0
8	1	0	0	1	1
9	1	0	1	1	0
10	0	1	1	1	0
11	1	0	1	1	0
12	0	1	0	0	1
13	1	0	0	0	1
14	0	1	1	1	0
15	0	0	1	1	0
16	1	1	1	0	0
17	1	0	1	1	0
18	0	1	0	0	1
19	0	1	0	0	1
20	1	0	0	0	1

**Table 2 healthcare-11-01917-t002:** Presence and testing strategies obtained for the example illustrated in Figure 1. The left binary digit is a presence indicator, present at the workplace (1), or works from home (0). The right binary digit shows he/she performs a test (1) or not (0). Note that it is possible an employee stays at home but does a test. This presence and testing strategy results in the expected risk of infection $1.99\times 10^{-5}$.

Employee	Monday	Tuesday	Wednesday	Thursday	Friday
1	0 ; 1	1 ; 0	1 ; 0	1 ; 1	0 ; 0
2	0 ; 1	1 ; 0	1 ; 0	1 ; 1	0 ; 0
3	0 ; 1	1 ; 0	1 ; 0	1 ; 1	0 ; 0
4	1 ; 1	0 ; 1	0 ; 0	1 ; 0	1 ; 0
5	1 ; 0	0 ; 1	0 ; 1	0 ; 0	1 ; 0
6	1 ; 0	1 ; 1	0 ; 0	0 ; 0	0 ; 1
7	1 ; 0	0 ; 1	0 ; 1	1 ; 0	1 ; 0
8	1 ; 1	0 ; 1	0 ; 0	0 ; 0	1 ; 0
9	0 ; 1	1 ; 0	0 ; 1	1 ; 0	1 ; 0
10	0 ; 1	1 ; 0	0 ; 1	0 ; 0	1 ; 0
11	0 ; 1	1 ; 0	0 ; 1	1 ; 0	1 ; 0
12	1 ; 1	0 ; 1	1 ; 0	1 ; 0	0 ; 0
13	0 ; 1	1 ; 0	1 ; 1	0 ; 0	0 ; 0
14	1 ; 0	0 ; 1	0 ; 1	1 ; 0	1 ; 0
15	0 ; 0	0 ; 1	1 ; 1	1 ; 0	0 ; 0
16	0 ; 1	1 ; 0	1 ; 1	1 ; 0	1 ; 0
17	1 ; 1	0 ; 1	1 ; 0	0 ; 0	0 ; 0
18	1 ; 0	0 ; 1	1 ; 1	0 ; 0	0 ; 0
19	1 ; 0	0 ; 0	0 ; 1	0 ; 1	1 ; 0
20	0 ; 1	1 ; 1	1 ; 0	0 ; 0	0 ; 0

**Table 3 healthcare-11-01917-t003:** Results obtained for the real data under 12 different scenarios. The reported risk values are scaled in $10^{5}$. For each scenario, three strategies are computed: (i) a feasible random strategy, denoted by column *R*. This solution satisfies all the constraints and is random in terms of both presence and testing strategies. (ii) The solution for the second model, denoted by M2, is for the case that the employees follow the suggested presence strategy with a random testing strategy. (iii) The solution for the first model, denoted by M1, for the case that the employees follow suggested strategies for both presence and testing.

Occupancy	TC	Min Presence: 2 Days			Min Presence: 3 Days		
		R	M2	M1	R	M2	M1
	**1**	5.81	4.31	2.38	9.51	7.97	4.40
[30%, 70%]	**2**	4.47	3.02	1.47	7.54	6.00	2.99
	**3**	3.31	1.96	1.10	5.77	4.11	2.27
	**1**	6.68	4.90	2.81	9.92	7.88	4.66
[40%, 80%]	**2**	5.12	3.51	1.93	7.60	5.97	2.90
	**3**	4.04	2.57	1.53	6.20	4.01	2.32

**Table 4 healthcare-11-01917-t004:** Results obtained for sparse random graphs. The reported risk values are scaled in $10^{5}$, and R,M2,M1 and TC are the same as explained in the previous results. Here, *n* is the number of nodes and FN is the probability of false negativity for each test.

n	FN	TC = 1			TC = 2			TC = 3		
		R	M2	M1	R	M2	M1	R	M2	M1
**40**	**0.3**	9.69	7.36	4.31	7.22	6.02	3.17	5.85	4.62	2.71
**40**	**0.1**	8.19	6.98	2.85	6.49	5.22	1.91	5.19	3.59	1.6
**100**	**0.3**	10.91	9.13	5.89	8.85	7.36	4.43	7.21	5.58	3.6
**100**	**0.1**	10.08	8.59	4.4	8.08	6.35	2.78	6.27	4.2	2.25
**250**	**0.3**	13.25	11.53	8.54	10.49	8.89	6.22	8.5	6.56	4.53
**250**	**0.1**	12.59	10.62	6.8	9.27	7.68	4.13	7.05	4.96	2.81

**Table 5 healthcare-11-01917-t005:** Results obtained for dense random graphs. The reported risk values are scaled in $10^{5}$, and R,M2,M1 and TC are the same as explained in the previous results. Here, *n* is the number of nodes and FN is the probability of false negativity for each test.

n	FN	TC = 1			TC = 2			TC = 3		
		R	M2	M1	R	M2	M1	R	M2	M1
**40**	**0.3**	13.42	11.50	6.38	11.52	9.27	4.68	9.28	6.99	3.89
**40**	**0.1**	12.77	10.86	4.19	9.76	7.97	2.74	8.28	5.26	2.10
**100**	**0.3**	16.39	11.78	7.30	11.53	9.27	5.41	9.41	6.93	4.33
**100**	**0.1**	14.65	11.00	5.47	10.08	7.96	3.49	7.81	5.24	2.44
**250**	**0.3**	24.17	20.28	14.24	18.67	14.93	9.54	13.77	10.6	6.78
**250**	**0.1**	23.08	18.55	11.02	17.16	12.07	5.38	11.14	8.31	5.35

**Table 6 healthcare-11-01917-t006:** Comparison results between APOPT solver and GA. The results are the average of 30 runs. The running times are in seconds, and the objective values are scaled in a factor of $10^{5}$.

*n*	Running Time		APOPT Solver			Genetic Algorithm		
	APOPT	GA	Min	Mean	Max	Min	Mean	Max
10	1.95	18.86	2.15	2.70	3.31	2.15	2.69	3.30
20	8.97	60.93	2.19	2.48	2.66	2.23	2.46	2.75
40	28.66	394	2.31	2.59	2.52	2.36	2.51	2.87
100	924	2291	2.53	2.82	3.87	2.53	2.72	3.18

**Table 7 healthcare-11-01917-t007:** Comparison results between Gurobi solver and GA. The results are the average of 30 runs. The running times are in seconds, and the objective values are scaled in a factor of $10^{5}$.

*n*	Running Time		Gurobi Solver			Genetic Algorithm		
	Gurobi	GA	Min	Mean	Max	Min	Mean	Max
10	3.06	14.92	5.04	5.83	6.28	5.04	5.76	5.92
20	12.98	52.35	5.08	6.16	6.54	5.08	5.68	6.44
40	0.55	322	5.28	5.62	6.94	5.17	5.52	6.82
100	0.34	1882	5.47	7.37	8.16	5.42	6.74	7.95

## Data Availability

The data in this research are the network of contacts among the employees depicted at https://www.sociopatterns.org (accessed on 15 June 2022).

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
