# Peer review of "Personnel Scheduling during the COVID-19 Pandemic: A Probabilistic Graph-Based Approach"

_healthcare, 2023, doi:10.3390/healthcare11131917_

Round 1
Reviewer 1 Report
This paper deals with personnel scheduling during the COVID-19 pandemic. The subject of the manuscript fits the Healthcare Journal. Figures and tables are present, they are clear and have adequate captions. References are proper and minimally adequate in quality and quantity; some adjustments may seem necessary for representativeness. The approach, organization, and results of the manuscript need minor adjustments. The document's English might need some minor grammar and style adjustments, and fixing typos may be necessary.
Major points to revise:
- Novelty needs to be highlighted in the Introduction;
- A framework or flowchart for the study, is necessary. There is a need for a detailed presentation of the steps performed aiming at the methodology approach's replicability and/or reproducibility. It aims at developing the study performed allowing a full understanding of the method and the proposed global validation, which are not clear yet;
- A better and more well-developed discussion is necessary. Although some validation is presented it needs further improvements for clarity and comprehensiveness. Further discussion of the results with a comparison with a proper literature review is necessary. Proper geographical distribution of the references will broaden the study and extend it to other places. Another point to bring attention to is the response to the questions raised that need to be clear and discussed;
- Some explanation of the assumptions made (lines 150 onwards) would clarify the approach to the solution;
- As mentioned, validation for the study performed is needed for clarification, the discussion part could reference the hypotheses and assumptions made;
- Indication of how practical and theoretical implications, with relevant insights, can be achieved, presented, and connected to the study field is interesting and can improve the paper usefulness.
The English is fine but one recommend that fixing typos and minor style and grammar issues is appropriate.
Reviewer 2 Report
This manuscript presented a COVID-19 two probabilistic model; the first model aims to derive optimal staff occupancy and testing strategies to minimize the risk of infection among employees, while the second model aims at only optimal staff occupancy under a random testing ten strategy. The topic is important and the manuscript is well-detailed; however, some concerns must be considered before it is accepted for publishing.
General Comments:
Although the topic of this article falls within the scope of the Journal, the authors have approached it in a highly technical way. They have deviated from the classical structure of the medical article. While I understand that the research problem requires mathematical presentation, I suggest modifying the paper to make it more accessible to clinicians and health specialists. Here are some suggestions that may benefit the authors:
Main issue:
The article is quite complex with several sections, which can make it hard to understand. However, it can be simplified as follows: Firstly, the introduction highlights the research problem and related work. Second, the model including its details such as parameters, variables, and assumptions. Thirdly, the model application, followed by a clear discussion. Finally, the conclusions are drawn, which include recommendations and implications for public health.
Specific Comments:
1. Abstract: this needs improvement as the focus is intensively on the research topic with no information about the input the model variables', primary outcome and health implications.
2. The introduction is not well prepared properly and need to be improved and the authors should
- The context of this study is limited to COVID-19 and not contribute much to public health management today. Please clarify the relevance of this study in the current background
- The introduction should better explain the purpose and motivation behind the study and its model design.
- Although several facts are mentioned, there are no citations provided in this section.
- Several parts of the introduction appear to overlap with the model section between lines 40-50.
- It would be beneficial to include a clear statement in the introduction section that highlights the contributions of this work to the existing literature.
3. The model and applications
- Model parameters (fully vaccinated) need to be clarified. Do you mean three doses of vaccine? This should be clearer for the reader.
- Also the rate transmission rate of vaccinated may differ according to the type of vaccine, please clarify?
- Is there any reference for FN =0.1, 0.2 and 0.3?
- Discussion: The results of Empirical Study have not been discussed for example models under various scenarios can reduce the risk of infection by 25% to 60%.
- What about the model limitations?
4. Conclusion:
The conclusion should be revised to focus on the model's outcome and recommendations. It's important to discuss the study's implications in the conclusion.
Round 2
Reviewer 2 Report
The paper has been improved significantly. I have no other issues.
Wishing the readers all the best with this wonderful manuscript.l manuscript.l manuscript.
Author Response
Thanks again for your help to improve the manuscript.